# Structural and compositional dependence of the $CdTe_xSe_{1-x}$ alloy layer photoactivity in CdTe-based solar cells

Jonathan D. Poplawsky[1], Wei Guo[1], Naba Paudel[2], Amy Ng[3,†], Karren More[1], Donovan Leonard[1] & Yanfa Yan[2]

The published external quantum efficiency data of the world-record CdTe solar cell suggests that the device uses bandgap engineering, most likely with a $CdTe_xSe_{1-x}$ alloy layer to increase the short-circuit current and overall device efficiency. Here atom probe tomography, transmission electron microscopy and electron beam-induced current are used to clarify the dependence of Se content on the photoactive properties of $CdTe_xSe_{1-x}$ alloy layers in bandgap-graded CdTe solar cells. Four solar cells were prepared with 50, 100, 200 and 400 nm-thick CdSe layers to reveal the formation, growth, composition, structure and photoactivity of the $CdTe_xSe_{1-x}$ alloy with respect to the degree of Se diffusion. The results show that the $CdTe_xSe_{1-x}$ layer photoactivity is highly dependent on the crystalline structure of the alloy (zincblende versus wurtzite), which is also dependent on the Se and Te concentrations.

[1] Center for Nanophase Materials Sciences and Department of Physics and Astronomy, Oak Ridge National Laboratory, Oak Ridge, Tennessee, 37831, USA. [2] Department of Physics and Astronomy, The University of Toledo, McMaster Hall, 2nd Floor Room 2017, Toledo, Ohio 43606, USA. [3] Vanderbilt University, Department of Chemistry, 7330 Stevenson Center, Nashville, Tennessee 37235, USA. † Present address: US Naval Research Laboratory, Materials Science & Technology Division, 4555 Overlook Avenue SW, Washington DC 20375, USA. Correspondence and requests for materials should be addressed to J.D.P. (email: poplawskyjd@ornl.gov).

Thin-film CdTe-based solar modules are relatively easy to produce at a low cost, making them one of the most competitive commercially available solar technologies in terms of price per watt, even though the world-record cell has a power conversion efficiency (PCE) of 22.1%, which is well below the theoretical PCE ($\sim$33%)[1]. An increase in the solar cell PCE without increasing the production costs will make solar power generation more competitive with fossil fuels, and the world's electricity needs can be fulfilled without significantly increasing the carbon footprint. The PCE record of CdTe-based solar cells remained relatively stagnant for roughly 20 years until several recent PCE enhancements were achieved, where the world-record cell PCE increased from 17.3% in 2012 to 21.5% in 2015 (refs 1,2). The latest increases in PCE are mostly attributed to an increase in the short-circuit current ($J_{sc}$)[1,2]. A comparison of the external quantum efficiency (EQE) measurements of Paudel et al.[3] to the world-record CdTe solar cell, together with the existence of a patent filed by First Solar (fabricator of the world-record cell), suggests that bandgap engineering using Se diffusion is one factor that enabled the $J_{sc}$ increases of the world-record cell over the past 4 years[1,3,4]. Paudel et al. showed that replacing the traditionally used CdS window layer with CdSe increases the $J_{sc}$ by increasing the cell photo-response for short- and long-wavelength photons. In addition, Yang et al.[5,6] have shown that using CdS/CdSe window layer stacks in CdTe solar cells increases the PCE compared with traditional CdS window layers. A detailed microscopic understanding in the growth, formation and photoactivity of the CdTe$_x$Se$_{1-x}$ alloy layer formed when CdTe is deposited on CdSe is necessary to optimize the fabrication of Se-induced bandgap-graded CdTe solar cells and further increase solar property parameters, such as the open-circuit voltage ($V_{oc}$), fill factor (FF) and the overall PCE of CdTe solar cells using Se diffusion.

The solubility of Se into CdTe is much higher than S, and, therefore, can easily diffuse from the CdSe layer into the CdTe during the CdTe growth and post-processing treatments[5–10]. The Se diffusion is expected to form a graded CdTe$_x$Se$_{1-x}$ layer that reduces the bandgap of CdTe due to bowing effects, which explains the increased photo-response for long-wavelength photons. The reported increase of photocurrent in the short-wavelength regime suggests that the deposited CdSe layer completely diffuses into the CdTe because residual CdSe, or CdS for CdS window layer CdTe solar cells, absorb photons without contributing to the photocurrent[3,11–13]. As Se and Te interdiffuse between CdTe and CdSe grains during the CdTe growth, a CdSe/CdTe pseudobinary system is formed.

Previous reports have shown that for $0.6 \leq x \leq 1$, the CdTe$_x$Se$_{1-x}$ alloy has a cubic zincblende structure and for $0 \geq x > 0.3$, the

CdTe$_x$Se$_{1-x}$ alloy has a wurtzite structure[14–17]. A gradient in the Se content is necessary to create a bandgap gradient, and, therefore, the structure, composition and photoactivity relationships of CdTe$_x$Se$_{1-x}$ alloys must be understood to improve devices using Se diffusion in CdTe solar cells.

In this study, the Se content and crystalline structures (zincblende versus wurtzite) of the CdTe$_x$Se$_{1-x}$ diffusion layers formed during the CdTe growth processes are directly correlated with the photoactivity and junction position to better understand the photovoltaic properties of the resultant CdTe$_x$Se$_{1-x}$ alloy layer. The carrier separation properties of CdSe/CdTe solar cells prepared with varied CdSe layer thicknesses (50, 100, 200 and 400 nm) are measured with 30–50 nm resolution using scanning electron microscopy (SEM)-based electron beam-induced current (EBIC), while the crystallography and Se concentration are measured at the nanoscale using transmission electron microscopy selected area diffraction (TEM-SAD) and atom probe tomography (APT), respectively. These complementary techniques reveal the dependence of the Se content and crystalline structure of the CdTe$_x$Se$_{1-x}$ layer during the CdTe growth processes on the ability of the CdTe$_x$Se$_{1-x}$ layer to convert photons into usable electrical power.

## Results

**Photovoltaic properties.** The EQE and $J$–$V$ curves acquired for the CdTe solar cells prepared with 50, 100, 200 and 400 nm CdSe window layers used in the microscopy studies along with data acquired for a 130 nm-thick CdS layer CdTe solar cell are shown in Fig. 1. The performance parameters ($V_{oc}$, $J_{sc}$, FF and PCE) of the representative solar cells can be found in Table 1. As shown previously by Paudel et al.[3], the cell with the 100 nm CdSe window layer had the highest $V_{oc}$, $J_{sc}$ and total PCE among the four fabricated CdSe window layer cells. Compared with the standard CdS/CdTe solar cell, the CdSe/CdTe cells showed an enhanced photo-response in the short- and long-wavelength regions when CdSe window layers were limited to <200 nm in thickness. However, the EQE in the short-wavelength regime drops markedly when the CdSe window layer thickness increases to 400 nm (Fig. 1b), causing a drastic decrease in the $J_{sc}$ (Table 1). A more detailed review of the solar cell properties, including a statistical analysis, with respect to the CdSe thickness and the comparison of an optimized CdSe window layer cell with a control CdS window layer cell, which yielded similar PCEs of approximately 15%, can be found in ref. 3.

The decreased EQE for the short-wavelength photons for the 400 nm CdSe layer cell suggests that a large part of the CdSe window layer interfacing the fluorine-doped-SnO$_2$/SnO$_2$ (FTO)

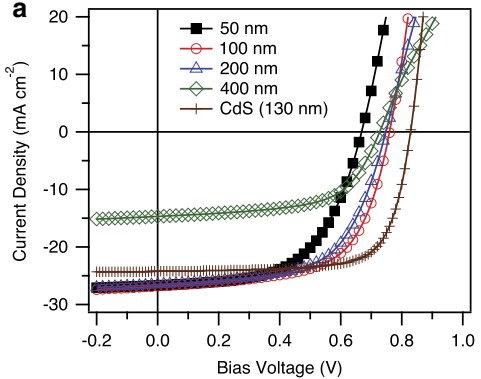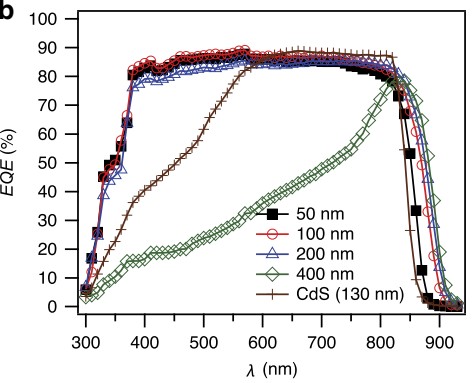

**Figure 1 | Photovoltaic properties.** (**a**) $J$–$V$ curves and (**b**) EQE measurements for CdSe layer thicknesses of 50, 100, 200 and 400 nm, and a CdS window layer cell with a 130 nm-thick CdS layer.

**Table 1 | Photovoltaic performance parameters.**

| CdSe layer Thickness | $V_{oc}$ (mV) | $J_{sc}$ (mA cm$^{-2}$) | FF (%) | PCE (%) |
|---|---|---|---|---|
| 50 nm | 670 | 26.7 | 57.0 | 10.2 |
| 100 nm | 770 | 27.0 | 60.2 | 12.6 |
| 200 nm | 748 | 26.5 | 60.5 | 11.9 |
| 400 nm | 739 | 14.7 | 61.2 | 6.6 |
| 130 nm (CdS) | 810 | 23.8 | 75.4 | 14.5 |

FF, fill factor; PCE, power conversion efficiency.
$V_{oc}$, $J_{sc}$, FF and PCE of the devices used in this study for the CdTe solar cells with different CdSe layer thicknesses.

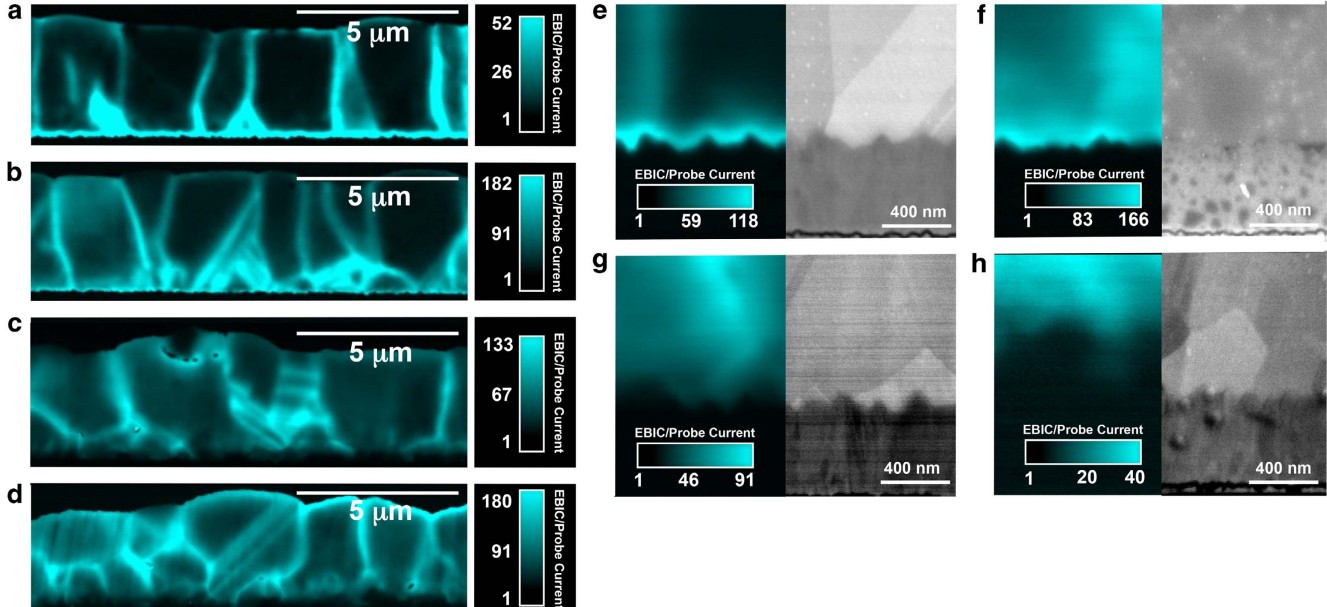

**Figure 2 | Cross-sectional EBIC maps.** (a–h) Cross-sectional EBIC maps of the devices prepared with (**a**) 50, (**b**) 100, (**c**) 200 and (**d**) 400 nm-thick CdSe layers. The associated contrast bars represent the EBIC current divided by the probe current (no units). SE-SEM images and simultaneously acquired EBIC maps of the devices with (**e**) 50, (**f**) 100, (**g**) 200 and (**h**) 400 nm CdSe layers. The 50 and 100 nm CdSe layers do not have small CdSe grains interfacing the FTO, while the 400 nm sample shows small CdSe grains interfacing the FTO. The EBIC map combined with the SE-SEM image for the 400 nm CdSe layer sample (**h**) shows that the small CdSe grains interfacing the FTO are not photoactive.

substrate is not photoactive, which would mean that the CdSe layer is not photoactive. However, the samples with 50–200 nm CdSe window layers do not exhibit drastic current losses for short-wavelength photons.

**Electron beam-induced current**. Figure 2a–d shows the cross-sectional EBIC maps for the 50, 100, 200 and 400 nm CdSe window layer samples, respectively. The 50 nm CdSe window layer cell sample (Fig. 2a) shows high EBIC signals within the grain boundaries (GBs) and at the CdSe/CdTe interface where space charge regions exist; however, the magnitude of the measured EBIC throughout the cell, especially the grain interiors, is rather low compared with the other samples, which correlates with a device $V_{oc}$ that is low[18–20]. The 50 nm CdSe layer does not form a good heterojunction with the CdTe and the device shares similar properties, such as $V_{oc}$, $J_{sc}$ and so on, with a window-less CdTe solar cell (CdTe directly deposited on FTO)[3]. The EBIC signal magnitudes throughout the active region, including the CdTe GBs, the CdSe/CdTe interface and the CdTe grain interiors, increase as the CdSe layer thickness is increased to 100 nm (Fig. 2b), which correlates well with the comparatively higher $V_{oc}$. The CdSe/CdTe interface does not show an enhanced EBIC signal

as the initial CdSe layer thickens (200 nm, Fig. 2c), and the average EBIC signals are reduced throughout the cell compared with the 100 nm-thick CdSe layer cell. These observations are corroborated by the reduction in $J_{sc}$, $V_{oc}$ and PCE for the 200 nm-thick CdSe layer compared with the 100 nm-thick layer device.

The most marked reduction in the overall PCE occurs for the 400 nm-thick CdSe layer (Fig. 2d), where $J_{sc}$ is nearly half that of the three thinner CdSe layer devices. The EBIC measurements show high EBIC signals across the cell; however, the front of the CdSe layer interfacing the FTO is not producing a measureable EBIC signal. Secondary electron (SE)-SEM images of the magnified FTO/CdSe interfaces and the associated EBIC maps are shown in Fig. 2e–h for the 50, 100, 200 and 400 nm-thick CdSe layers, respectively. The entire CdSe layer is EBIC active, and, hence, photoactive up to the FTO interface for the 50 and 100 nm-thick CdSe layers, whereas the 200 nm CdSe layer shows regions with reduced EBIC signals close to the interface and the 400 nm layer shows an approximately 400 nm layer adjacent to the FTO that does not produce an EBIC signal, which indicates that this layer is not photoactive.

Average EBIC line profiles of the four samples prepared with different CdSe thicknesses are shown in Fig. 3 and clarify the location of the pn-junction with respect to the initial CdSe layer

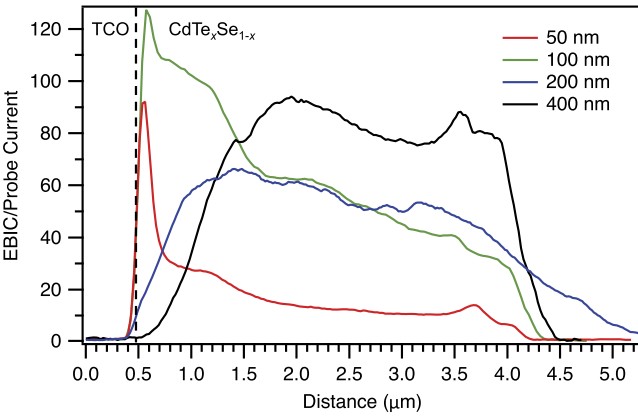

**Figure 3 | Cross-sectional EBIC average line profiles.** EBIC average line profiles of the devices with 50, 100, 200 and 400 nm-thick CdSe layers with the approximate location of the FTO/CdTe$_x$Se$_{1-x}$ interface marked by the dashed black line. Note that the roughness of the FTO layer and the electron beam excitation volume create some error in the exact location of the FTO/CdTe$_x$Se$_{1-x}$ interface.

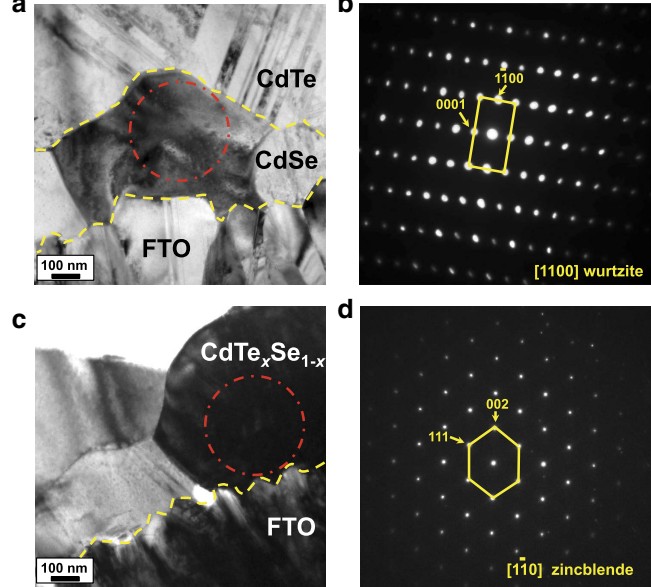

**Figure 4 | CdTe growth-induced phase transformation.** (**a–d**) BF-TEM images and SAD patterns of structures associated with the FTO/CdSe/CdTe interfaces. The yellow-dashed lines designate the FTO, CdSe and CdTe interfaces for each sample. (**a**) BF-TEM image of the FTO/CdSe/CdTe structure for the cell with a 400 nm-thick CdSe layer. (**b**) TEM-SAD pattern acquired from the grain region outlined with a red dashed circle in **a**. The SAD pattern is indexed as [1100] for the CdSe wurtzite phase. (**c**) BF-TEM image of the FTO/CdTe$_x$Se$_{1-x}$ region for the cell with a 100 nm-thick CdSe layer. (**d**) TEM-SAD pattern acquired from the grain region outlined with a red dashed circle in **c**. The SAD pattern is indexed as [1̄10] for the CdTe$_x$Se$_{1-x}$ zincblende phase.

thickness. The 50 and 100 nm CdSe layer samples show a junction that is very close to the FTO/CdTe$_x$Se$_{1-x}$ interface, such that the EBIC measurements do not discern whether the pn-junction occurs directly at the FTO/CdTe$_x$Se$_{1-x}$ interface or slightly within the CdTe$_x$Se$_{1-x}$ alloy layer close to the interface. The increased $V_{oc}$ of the cell with the 100 nm CdSe layer compared with the 50 nm CdSe layer suggests that less recombination is occurring at the FTO/CdTe$_x$Se$_{1-x}$ interface, such that the additional CdSe better passivates the surface between the FTO and CdTe$_x$Se$_{1-x}$ layers. The EBIC average line profiles clearly show that the junction is beyond the FTO/CdTe$_x$Se$_{1-x}$ interface, within the CdTe$_x$Se$_{1-x}$ layer, for the 200 and 400 nm samples, and that the final junction position (distance from the original FTO/CdSe interface) is dependent on the original CdSe layer thickness.

The grains interfacing the FTO layer resemble large columnar and twinned CdTe grains, as shown by the SE-SEM images in Fig. 2e–h. These grains extend all the way to the back contact with the 100 nm CdSe layer (Supplementary Fig. 1). This is preliminary evidence that the original CdSe layer alloys and reacts with the CdTe and recrystallizes, likely during both the CdTe growth procedure and subsequent heat treatments. The SEM images show that the microstructure of the cell with the 400 nm CdSe layer is quite different, with smaller grains interfacing the FTO layer that do not extend to the back contact (Fig. 2h) and larger columnar CdTe-like grains above these smaller grains (Supplementary Fig. 1), which could indicate the presence of residual (unreacted) CdSe grains at the FTO/CdSe interface. EBIC data show that the small grains interfacing the FTO layer are not photoactive, and, thus, TEM-SAD was used to identify the crystalline phases present close to the FTO/CdTe$_x$Se$_{1-x}$ interface for the cells with 100 and 400 nm-thick CdSe layers.

**Transmission electron microscopy.** To elucidate the origin of the change in photoactive properties for the cells with 100 and 400 nm CdSe layers, focused ion beam (FIB) cross-sections were prepared for TEM analysis across the FTO/CdTe$_x$Se$_{1-x}$ interfaces. A bright-field (BF)-TEM image of the 400 nm CdSe layer is shown in Fig. 4a. The two yellow-dashed lines designate the boundary or interface between different constituent layers. The grains in the top most layer contain a high density of twins[21], which are typical of cubic zincblende CdTe grains grown by the

closed space sublimation (CSS) process. The red circle shown within the dark grain highlights the region of interest (ROI) selected for SAD. The indexed TEM-SAD pattern shown in Fig. 4b is consistent with the CdSe hexagonal wurtzite structure with the grain aligned along the [1100] zone axis ($d_{(1\bar{1}00)} = 3.723$ Å, $d_{(0001)} = 7.179$ Å). The measured lattice parameters for this CdSe grain are $a_0 = 4.3$ Å and $c_0 = 7.179$ Å, compared with the literature values that range between $a_0 = 4.3$ Å–4.4 Å and $c_0 = 7.01$ Å–7.2 Å for CdSe with a relatively high Se content ($>35\%$)[22,23]. This result demonstrates that the original deposited 400 nm CdSe layer retains the wurtzite structure during CSS CdTe growth and subsequent heat treatments.

A BF-TEM image of the 100 nm CdSe layer is shown in Fig. 4c, which shows no clear boundary between the CdSe and CdTe layers as observed for the cell with the 400 nm CdSe. This is consistent with the SE-SEM images shown in Supplementary Fig. 1, where columnar CdSe grains extend the entire distance between the FTO layer and the back contact. The TEM-SAD pattern shown in Fig. 4d acquired from a single-grain adjacent to the FTO (red circle in Fig. 4c) further confirms this observation. The SAD pattern from this grain was oriented along the [1̄10] zone axis and was indexed as the CdTe$_x$Se$_{1-x}$ zincblende structure. The measured lattice parameter for this CdTe$_x$Se$_{1-x}$ grain was $a_0 = 6.480$ Å, which is consistent with the 6.480–6.483 Å literature values[24–26]. The phase diagram for CdTe$_x$Se$_{1-x}$ alloys indicates that a threshold value of $x$ ranging from 0.4 to 0.55 enables a phase transformation from the CdSe wurtzite structure to the CdTe$_x$Se$_{1-x}$ zincblende structure[16,23]. Thus, for the cell with a 100 nm CdSe layer, the original CdSe grains reacted with the CdTe during growth with significant Te and Se

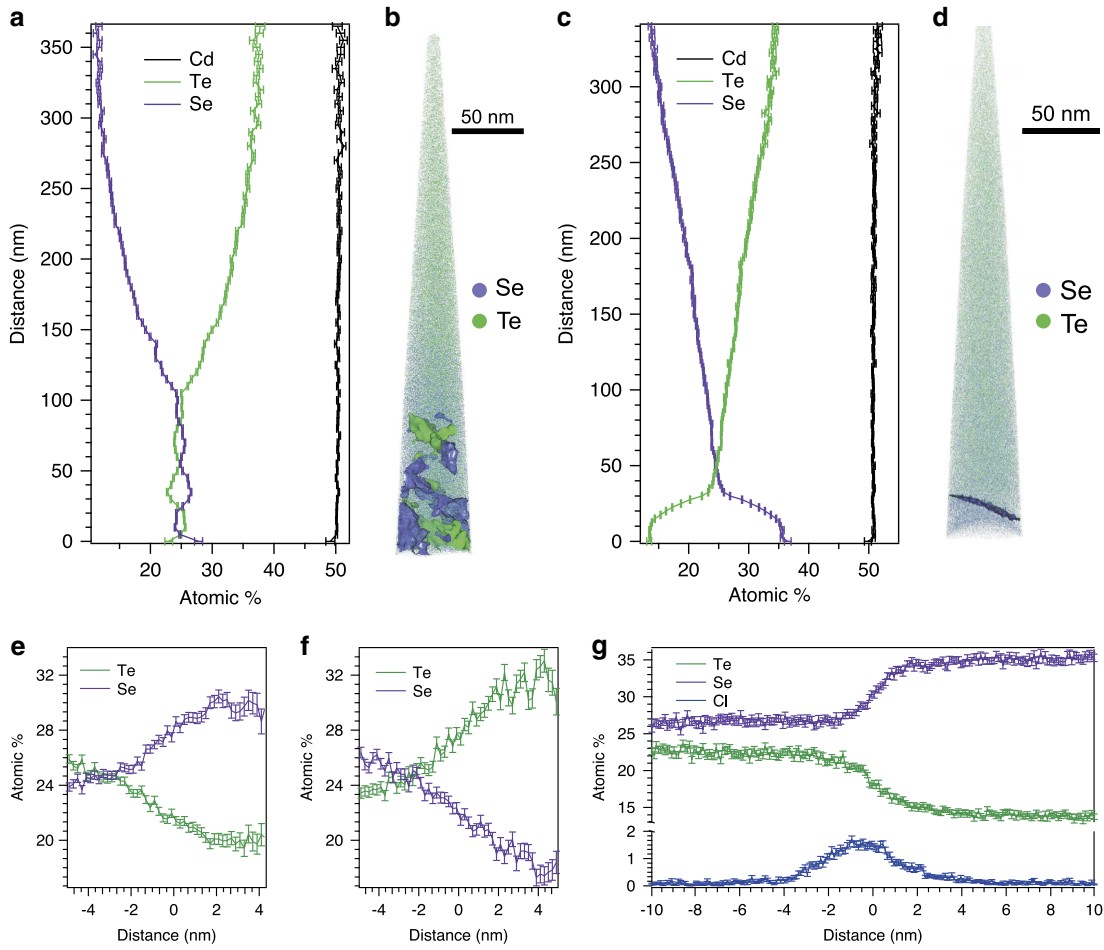

**Figure 5 | Compositional variations at the nanoscale.** (**a**) A 1D line profile in the *z* direction of the (**b**) APT 3D reconstructed data set for the 100 nm CdSe layer sample displaying the Te (green) and Se (purple) atoms, and 27.5% Te (green) and Se (purple) isosurfaces. (**c**) A 1D line profile in the *z* direction of the (**d**) APT 3D reconstructed data set for the 400 nm CdSe layer sample displaying the Te (green) and Se (purple) atoms, and a 24% Se isosurface. Proximity histograms for the 27.5% Te (**e**) and Se (**f**) isosurfaces for the 100 nm sample imaged in **b**. (**g**) Proximity histogram for the 24% Se isosurface shown in **d**. The error bars embedded in the 1D line profiles and proximity histograms are the s.e. (*s*) from counting statistics, $s = \sqrt{\frac{c(1-c)}{N}}$, where *c* is the concentration of the element and *N* is the total number of ions. Minor uncertainties can occur from background levels and mass spectral peak overlaps. A detailed summary of the mass spectrum, mass spectral ranges and resulting compositions can be found in Supplementary Fig. 4 and Supplementary Table 2.

interdiffusion occurring at the CdSe/CdTe interface forming a $CdTe_xSe_{1-x}$ alloy. The Se content of this alloy must have reduced to below 35 at. % to enable the phase transition from photoinactive wurtzite CdSe to the photoactive $CdTe_xSe_{1-x}$ zincblende phase. During the phase transformation as a result of the Se/Te interdiffusion, the originally deposited CdSe grains recrystallized with the same orientation as the deposited CdTe grain, forming columnar grains extending from the FTO to the back contact. Such a phase transformation allows for the 100 nm sample to be photoactive across the entire $CdTe_xSe_{1-x}$ layer, while the large columnar grains extending from the back contact are terminated by small non-photoactive wurtzite $CdTe_xSe_{1-x}$ grains for the 400 nm-thick CdSe layer. It can be expected that the 400 nm CdSe layer retained a higher Se content than the 100 nm CdSe layer if the same amount of Te and Se diffusion occurred at the CdSe/CdTe interface during the CdTe CSS growth process; APT was used to verify this hypothesis.

**Atom probe tomography.** APT is a popular technique for detecting compositional variations at the nanoscale for CdTe-based solar cells due to its high elemental sensitivity, spatial resolution and relative ease of running CdTe materials in APT

since the introduction of laser pulsing[27–34]. To further elucidate the dependence of stable phase formation within the CdSe/CdTe layers close to the FTO as a function of Se content, APT data were acquired for the $FTO/CdTe_xSe_{1-x}$ interfacial regions for the cells with 100 and 400 nm-thick CdSe layers.

Figure 5b,d shows the three-dimensional (3D) APT reconstructions from representative CdSe/CdTe interfacial volumes with their respective one-dimensional (1D) Cd, Se and Te concentration profiles in Figure 5a,c, for the 100 and 400 nm-thick CdSe layers, respectively. The 1D concentration profiles clearly show that the $CdTe_xSe_{1-x}$ alloy in both samples exhibit graded Se and Te concentrations with the Se-rich region at the front of the cell. This is direct evidence that these solar cells are bandgap graded because the bandgap of the $CdTe_xSe_{1-x}$ alloy is dependent on the value of *x*, where the bandgap of CdTe can decrease by as much as 8%, when *x* = 0.6. A detailed review of how bandgap energies change with respect to the $CdTe_xSe_{1-x}$ alloy composition can be found in ref. 12.

The reconstructed APT volumes shown in Fig. 5b,d are comprised of purple and green dots that represent the 3D locations of the Se and Te atoms, respectively. Data were acquired for the APT sample with the 100 nm CdSe layer across the entire

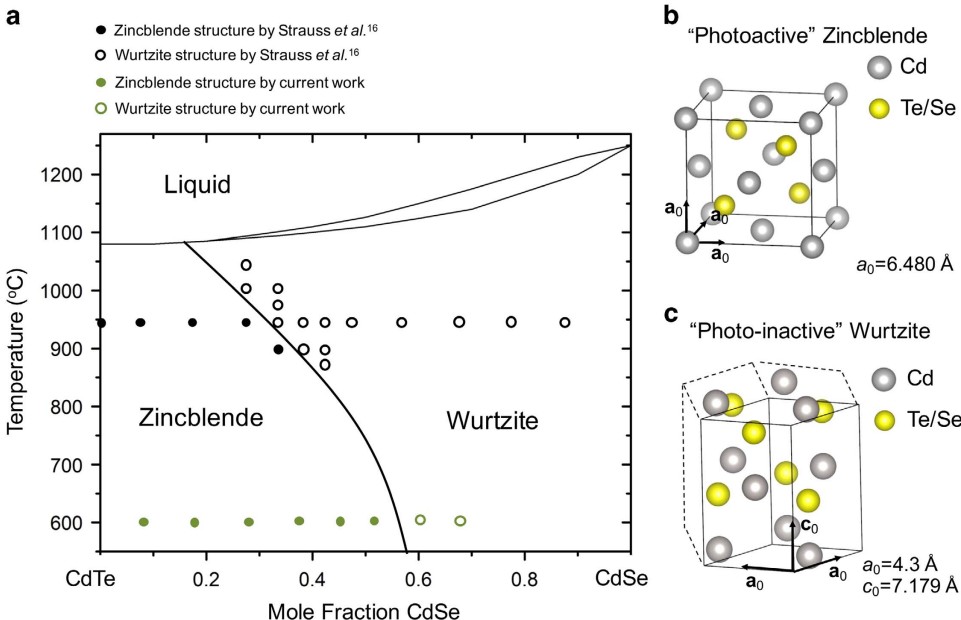

**Figure 6 | Phase diagram of the CdTe-CdSe pseudobinary system.** (**a**) An updated phase diagram for the CdSe-CdTe pseudobinary system using data from Strauss et al.[16] and the APT and TEM-SAD results from the present studies. (**b**) Schematic of the unit cell of the photoactive zincblende structure. (**c**) Schematic of the unit cell of the photoinactive wurtzite structure.

$CdTe_xSe_{1-x}$ layer and ended at the FTO layer, while the 400 nm sample fractured after passing through the interface separating the Se- and Te-rich phases. Unlike the data acquired for the sample with a 400 nm-thick CdSe layer, there is no distinct/rapid change in composition between the original CdSe and CdTe layers for the sample with the 100 nm CdSe layer. Rather, the Se and Te contents remain constant at ∼25 at. % between the FTO and 100 nm into the $CdTe_xSe_{1-x}$ alloy. Isoconcentration surfaces for 27.5 at. % Te (green) and 27.5 at. % Se (purple) are also included in the APT reconstructions (Fig. 5b,d), which separate regions of the data set that are >27.5% and <27.5% Te and Se, respectively (the interior of the green surfaces are Te-rich regions, while the interior of the purple surfaces are Se-rich regions)[35]. Proximity histograms of the Te and Se isosurfaces shown in Fig. 5b,d are shown in Fig. 5e,f, respectively, which show the Te and Se concentration profiles calculated with respect to distance from the isoconcentration surfaces[36]. Te contents as high as 32 at. % were identified inside the Te-rich regions and Se contents as high as 30 at. % were identified inside the Se-rich regions, while the Cd composition remained constant within the analysed volume. Scanning transmission electron microscopy (STEM)-based energy dispersive X-ray spectroscopy (EDS) confirmed a non-homogenous Se distribution close to the FTO interface and showed a graded Se distribution, with the highest Se concentration at the FTO interface that decreased with distance from the interface into the CdTe layer (Supplementary Fig. 2).

While TEM-SAD was used to verify that the large columnar grains adjacent to the FTO layer shown in Supplementary Fig. 1 were zincblende for the 100 nm CdSe layer sample, APT and STEM-EDS results showed that the composition within the grains was not constant. The fluctuating Te- and Se-rich regions associated with the location of the original CdSe layer (approximately 100 nm into the $CdTe_xSe_{1-x}$ alloy away from the FTO/$CdTe_xSe_{1-x}$ interface) may be a result of varied interdiffusion dynamics during the reaction between CdSe and CdTe, and the rather small CdSe grain size (<50 nm) and large network of GBs within the CdSe layer before CdTe deposition.

The sample with the 400 nm CdSe layer exhibited a clear interface between the Te-rich and Se-rich phases, such that a 24% Se isosurface could be used to define and separate two large, distinct $CdTe_xSe_{1-x}$ grains with significantly different Te and Se contents. A proximity histogram (Fig. 5g) across the isosurface shown in Fig. 5d reveals a rapid compositional change, where the Se content changes from 35 at. % to 25 at. % within a distance of only 5 nm, which is consistent with the measurements of S diffusion into CdTe grains for solar cells with CdS window layers[34,37]. In addition, Cl segregates between these separate alloy layers (Fig. 5g), which is consistent with Cl segregation at the CdS/CdTe interface for CdS window layer solar cells[18,34]. These compositional results combined with the structural data from TEM-SAD suggest that the interface defined by the isosurface in Fig. 5d is a wurtzite/zincblende interface within the $CdTe_xSe_{1-x}$ alloy layer, where regions having ≤25 at.% Se are identified as zincblende and ≥35 at.% Se are identified as wurtzite.

The 1D line profile shown in Fig. 5c shows an extended Se profile well past the interface, where the Se content approaches zero several micrometers away from the FTO interface (Supplementary Fig. 3; Supplementary Table 1). Se diffusion into the CdTe layer for CdSe window layers is of greater significance than S diffusion for CdS window layers because >25 at. % Se diffuses and incorporates into the zincblende CdTe grains, while only roughly 5 at. % S diffuses into the CdTe grains for cells with a CdS window layer exposed to the same growth conditions and experimental techniques[34,37].

The 1D APT line profiles for the samples with the 100 and 400 nm CdSe layers (Fig. 5a,c, respectively) show that significant Se diffuses into the deposited CdTe layer, forming distinct $CdTe_xSe_{1-x}$ alloy layers between the FTO and CdTe, and that the Se content decreases to much less than 1 at. % several micrometers away from the FTO interface. The compositional gradient is direct evidence for the existence of a bandgap gradient within the active layer of the device. However, the inherent wurtzite structure of CdSe is maintained when the Se content increases beyond 35 at.% Se, which is a non-photoactive

phase. These data explain the significantly reduced $J_{sc}$ for the device with a 400 nm CdSe layer.

**Updated CdTe$_x$Se$_{1-x}$ phase diagram.** Although both wurtzite and zincblende CdTe$_x$Se$_{1-x}$ structures exist with 30 at. % Se[23], the results presented here show that a zincblende CdTe$_x$Se$_{1-x}$ alloy with $\leq$ 30 at.% Se results from using the CSS growth technique and this layer is photoactive. Therefore, an updated phase diagram is presented that includes a combination of the data from Strauss et al.[16] and the correlation between the APT compositions and TEM-SAD structures from the present work, which is shown in Fig. 6. The black and white dots represent the zincblende and wurtzite structures, respectively, as suggested by Strauss et al.[16], and the green and white dots represent the zincblende and wurtzite structures, respectively, from the current work. The phase diagram presented by Strauss et al.[16] is limited to a minimum temperature of 800 °C, and, thus, the modified phase diagram is extended to the CdTe substrate growth temperature (610 °C). The two-phase data points that encompass a small fraction of the phase diagram from Strauss et al.[16] were left off the modified phase diagram to maintain simplicity. Strauss et al.[16] annealed the specimens from 16 h to 30 days, depending on the annealing temperature used, to reach the equilibrium state, and then quenched the samples in water. The phases were identified using X-ray diffraction. On the other hand, the CdTe$_x$Se$_{1-x}$ alloy used for the new data points added to the phase diagram from the present work was formed while CdTe was deposited on top of a thin CdSe layer at 610 °C for several minutes, followed by a 30 min 390 °C CdCl$_2$ anneal and a 30 min 150 °C Cu diffusion step. The specimen was not quenched after the anneal treatments.

## Discussion

The best CdSe/CdTe solar cells that have been produced using the growth method used for the cells studied in this manuscript have similar efficiencies to CdS/CdTe solar cells (14.7% versus 14.8% for the best cells)[3]. The PCE of the CdSe/CdTe solar cells can be improved by improving the $V_{oc}$ and FF, which were both lower than measured for the CdS/CdTe control cells. Time-resolved photoluminescence measurements showed a significantly lower carrier lifetime as compared with non-alloyed CdTe thin films, indicating that increased recombination reduces the $V_{oc}$ and FF of the studied CdTe$_x$Se$_{1-x}$ alloy layers. This may be due to the Se non-uniformity (shown by the APT results in Fig. 5a–d) that occurs as the CdSe and CdTe layers react and Se diffuses from the CdSe layer into the CdTe and Te diffuses from the CdTe into the CdSe layer. Future work should be aimed towards increasing the carrier lifetimes, $V_{oc}$, and FF by creating a more uniformly graded CdTe$_x$Se$_{1-x}$ alloy. A Se alloy source can potentially be used to improve the uniformity of the graded CdTe$_x$Se$_{1-x}$ alloy, and further improve the device PCE beyond a typical CdS/CdTe device. Also, the shorter minority carrier lifetimes of the CdSe/CdTe devices compared with that of CdS/CdTe can be due to interfacial recombination at the FTO/CdTe$_x$Se$_{1-x}$ alloy interface. Optimization of a buffer layer between the FTO and CdTe$_x$Se$_{1-x}$ alloy layers, such as CdS, can further increase the device PCE beyond CdTe solar cells that are not bandgap graded, as shown by Yang et al.[5]

Overall, the incorporation of a CdSe window layer increases the $J_{sc}$, long-wavelength absorption and short-wavelength absorption of CdTe solar cells. The nature of Se diffusion from a deposited CdSe layer during CSS growth of CdTe and subsequent heat treatments, and its effect on the photovoltaic properties of CdTe devices, have been studied using a combination of SEM-EBIC, TEM-SAD and APT. APT results show that Se diffuses

from the CdSe layer into the CdTe layer, forming a CdTe$_x$Se$_{1-x}$ alloy and a Se concentration gradient within the CdTe. When the original CdSe layer is thin enough to significantly react with CdTe within the FTO/CdTe interface, it fully transitions into a zincblende-structured CdTe$_x$Se$_{1-x}$ alloy that forms as large columnar grains that extend from the FTO to the back contact (shown by TEM and SEM imaging and TEM-SAD). SEM-EBIC measurements confirms that the zincblende CdTe$_x$Se$_{1-x}$ alloy layer is photoactive. For thicker CdSe layers (400 nm), the original CdSe layer does not incorporate a sufficient amount of Te for the complete wurtzite-to-zincblende phase transformation during CdTe growth and subsequent heat treatments, such that the CdTe$_x$Se$_{1-x}$ alloy layer contains both wurtzite and zincblende grains, with the higher Se content wurtzite grains interfacing with the FTO layer. SEM-EBIC data confirm that the wurtzite phase is photoinactive. EBIC average line profiles for all the devices with varying CdSe layer thicknesses show that the junction position is dependent on the original CdSe layer thickness. Overall, a photoactive layer containing a bandgap gradient, with a reduced bandgap close to the FTO/CdTe$_x$Se$_{1-x}$ interface, can be formed by diffusing an average of 25 at. % Se into the CdTe layer. This gradient increases the $J_{sc}$ by increasing the absorption of long-wavelength photons. A bandgap-graded absorber layer is one critical component for high PCE CdTe-based solar cells. Therefore, the optimization of Se diffusion into the CdTe absorber layer is crucial for improving the state of the art of CdTe-based solar cells.

## Methods

**CdTe solar cell synthesis and property measurements.** Four CdSe/CdTe cell structures with different starting CdSe layer thicknesses were fabricated. Each cell was grown on a commercial FTO soda lime glass supplied by NSG North America. The CdSe layers were magnetron sputtered on the FTO glass substrates at 250 °C using a 50 W radio frequency power in a 10 mTorr Ar purged chamber with CdSe layer thicknesses of 50, 100, 200 and 400 nm. Following the CdSe deposition on the FTO, each cell was exposed to the same growth and post-deposition procedures: a 4 µm CdTe layer was grown on the CdSe by CSS with a 610 °C substrate temperature, followed by a CdCl$_2$ heat treatment at 390 °C for 30 min, a Cu diffusion step and application of the back contact. Additional details of the growth procedure can be found elsewhere[3].

All $J$–$V$ curves were generated at room temperature in ambient air (25–30 °C) using a calibrated Xenon-arc lamp purchased from Newport with a reverse to forward bias (− 0.2 to + 1.0 V), 10 mV step size and 300 ms dwell time. National Renewable Energy Laboratory (NREL)-certified GaAs and Si reference cells were used to match the solar spectrum and 1-sun intensity. The cells were 0.08 cm$^2$ and were isolated using a shadow mask (0.08 cm$^2$) and scribing. The size of the shadow mask did not change the $J_{sc}$. The current density was also verified by the EQE measurements.

**Electron beam-induced current.** The SEM-EBIC measurements were performed in a Hitachi S4800 cold field emission gun SEM equipped with a Gatan SmartEBIC system. All EBIC specimens were cross-section argon ion milled at liquid nitrogen temperatures using a Gatan Ilion + ion polishing system to ensure a smooth, relatively damage-free surface. An Ar$^+$ beam acceleration voltage of 5 kV allowed for less than 10 nm of surface damage, as calculated from Stopping and Range of Ion in Matter simulations[38,39]. The cross-sectioned devices were mounted on the EBIC stage and connected to a Stanford Research Instruments SR570 low-noise current amplifier. A 3 kV accelerating voltage and beam current of 30–50 pA was used for SEM-EBIC to ensure a high-resolution probe, while maintaining low-injection conditions[40]. A low accelerating voltage ensures that the EBIC signal is sensitive to the magnitude of local electric fields such that space charge regions at the pn-junction and GBs can be easily imaged[41]. All EBIC measurements represent the microscopic short-circuit carrier extraction efficiencies because they were conducted without an applied bias.

**Atom probe tomography.** An FEI Nova 200 dual-beam FIB-SEM instrument was used to perform site-specific lift-outs of specimen ROIs and to perform annular milling to fabricate the needle-shaped APT specimens. A wedge lift-out geometry was used to mount multiple samples on a Si microtip coupon to enable the fabrication of multiple APT needles from one wedge lift-out[42]. A 5 kV final polishing step minimized Ga implantation damage and sharpened the APT needle within the specific ROI selected. The method used to prepare APT

specimens from the CdSe/CdTe interface required additional steps, which are detailed elsewhere[34].

The APT data were acquired using a local electrode atom probe (CAMECA Instruments LEAP 4000X HR) equipped with an energy compensated reflectron lens and a 36% detection efficiency. The APT experiments were conducted at a specimen base temperature of 30 K by applying 3 pJ, 355 nm wavelength, 10 ps laser pulses at a repetition rate of 100 kHz, and a detection rate of 0.5–1.0%. Data reconstruction and analyses were performed using CAMECA IVAS 3.6.8 software.

**Transmission electron microscopy.** TEM specimens less than 100 nm-thick were prepared using a Hitachi NB5000 FIB-SEM via a standard *in situ* lift-out method, which was followed by low-kV argon ion-milling using a Fischione 1040 Nanomill. An FEI Tecnai T20 TEM operated at 200 kV was used for BF-TEM imaging and SAD observations. STEM-EDS spectrum images were acquired using a Hitachi HF3300 TEM-STEM equipped with a Bruker X-Flash silicon drift detector EDS system and operated at 300 kV.

**Data availability.** The data that support the findings of this study are available from the corresponding author on request.

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

## Acknowledgements

This research was supported by the US Department of Energy (DOE) Office of Energy Efficiency and Renewable Energy, Foundational Program to Advance Cell Efficiency, grant number DE-FOA-0000492, and performed in part at ORNL's Center for Nanophase Materials Sciences, which is a DOE Office of Science User Facility. J.P. was supported in part by ORNL's Laboratory Directed Research and Development program. We thank Dorothy Coffey and Shawn Reeves for specimen preparation.

## Author contributions

J.P. and A.N. performed the EBIC measurements. J.P. and W.G. performed the APT and TEM-SAD measurements. N.P. and Y.Y. developed and fabricated the CdSe/CdTe solar cell devices. K.M. performed the STEM-EDS measurements. D.L. performed the transmission electron backscatter diffraction measurements of the CdSe/CdTe interface to corroborate the TEM-SAD measurements. This manuscript has been authored by UT-Battelle, LLC under contract no. DE-AC05-00OR22725 with the US Department of Energy (DOE). The United States Government retains and the publisher, by accepting the article for publication, acknowledges that the US Government retains a non-exclusive, paid-up, irrevocable, world-wide license to publish or reproduce the published form of this manuscript, or allow others to do so, for US Government purposes. The DOE will provide public access to these results of federally sponsored

research in accordance with the DOE Public Access Plan (http://energy.gov/downloads/doe-public-access-plan).

## Additional information

**Competing financial interests:** The authors declare no competing financial interests.

