## [Peer Review File · Nature Communications]

[Editorial note: Some references in the Transparent Peer Review File are cited using capitalized letters in brackets and the reference list is given at the end of the Transparent Peer Review File. Numbers refer to the reference lists in the article or its previous versions.]

Reviewers' comments:

Reviewer #1 (Remarks to the Author):

The key results from this paper are that Se diffuses from a CdSe layer and forms a CdTe_{1-x}Sex layer which is photoactive. I do not feel that CdSe increases in Jsc can be considered a result from this paper as it this has already been published by Paudel in previous papers, particularly reference 3.

This is in essence though a good piece of work. The microscopy and APT data is solid and EBIC analysis is interesting and correlates with the EQE analysis to explain the junction shift caused by thick CdSe layers. However I do not feel this is sufficient new data to warrant publication in a non-specific PV journal such as this. Furthermore I feel the inability of this work to demonstrate an improvement in performance will limit the impact and interest in this work.

The authors are very careful to phrase their discussion in terms of improving Jsc but this really is of little benefit if the Voc and FF levels cannot be at least maintained. As a result the CdSe incorporation has simply served to reduce efficiency. If increasing the Jsc alone is the sole aim why not simply use a smaller bandgap absorber to begin with? I understand the bandgap grading aims but until this can be demonstrated to improve performance rather than simply long wavelength response it is of little consequence. At a time when the majority of the research field are focussing on attempts to improve Voc, a process which actively decreases this will receive little uptake.

The claim that first solar use a CdTe_{1-x}Sex layer is also unsubstantiated and their bandgap grading (which maintains or improves Voc) could be achieved via a number of other routes. Whilst the microstructural analysis provided is of a good quality some demonstration of bandgap engineering occurring, as invoked in the abstract, would be hugely beneficial to the work. If the authors could show they had produced a significant amount of bandgap grading through their structure this would be a more key finding.

Overall I feel this paper should be published in a more PV specific journal.

The paper is extremely well written and I find few typographical errors. The conclusions are appropriate and supported by the data although I disagree with glossing over the efficiency decreases to focus on Jsc. I have no concerns with the accuracy or validity of the data and all data analysis has been done to a high standard. The methodologies and processes employed are excellent as one would expect from groups of such high quality.

Specific corrections:

Abstract: Surely the Voc rather than Jsc is the key to breaking 25% efficiency? First solar are putting in little effort to increase Jsc and their bandgap grading has been done to improve the Voc.

Figure 2a - Scale bars are missing, what is the colour scale bar units? Is this output current in pA or is it EBIC current/Probe current?

Figure 3 - Presumably this should be μm rather than nm ?

Reviewer #2 (Remarks to the Author):

Summary of the key results

Using CdSe instead of CdS at the front contact increases the J_{sc} and wavelength absorption both for short and long wavelengths. This is due to the diffusion of Se into the CdTe creating zincblende CdTe $_{1-x}$ Se $_x$. If the wurtzite structure of CdSe is retained, then the J_{sc} is diminished.

Originality and interest: if not novel, please give references

This is certainly of interest to those in the thin film solar cell community. Some of these claims are novel, some have recently been reported by others. The following references explore the use of CdSe in the window layer and all report improved J_{sc} and improved short and long wavelength performance. All of the references also all attribute the improvement to interdiffusion of the CdSe and CdTe to create a graded CdTe $_{1-x}$ Se $_x$ layer:

Paudel, Naba R., and Yanfa Yan. "Enhancing the photo-currents of CdTe thin-film solar cells in both short and long wavelength regions." *Applied Physics Letters* 105 (2014): 183510.

Paudel, Naba R., et al. "Current Enhancement of CdTe-Based Solar Cells." *IEEE Journal of Photovoltaics*, 5 (2015): 1492-1496.

Yang, Xiaoyan, et al. "Preparation and characterization of pulsed laser deposited a novel CdS/CdSe composite window layer for CdTe thin film solar cell." *Applied Surface Science* 367 (2016): 480-484.

Yang, Xiaoyan, et al. "Preparation and characterization of pulsed laser deposited CdS/CdSe bi-layer films for CdTe solar cell application." *Materials Science in Semiconductor Processing* 48 (2016): 27-32.

The latter two explore the phase somewhat using XRD, but not on the detailed local scale that is presented here. The present work is novel in that it quantitatively explores the amount of interdiffusion that can occur and examines the local crystal structure of the CdTe $_{1-x}$ Se $_x$ phase.

Data & methodology: validity of approach, quality of data, quality of presentation

The use of electron diffraction, APT, and EBIC to examine the structure, composition, and photoactivity relationships of CdTe $_{1-x}$ Se $_x$ alloys are quite appropriate and together present an overall picture of what is occurring.

Appropriate use of statistics and treatment of uncertainties

The only mention of uncertainties is in the supplementary discussion of the APT data. While a counting statistics approach is one possible way of assessing the uncertainty, for APT it really represents the lowest possible uncertainty. A mention of the observed background levels, method of peak ranging, and any peak overlaps which require deconvolution would provide a more complete assessment of the uncertainty and reliability of the data.

Conclusions: robustness, validity, reliability

This presents a consistent microscopic picture of the observed changes in device performance and puts forth reasonable discussion regarding the results.

Suggested improvements: experiments, data for possible revision

The two references by Yang et al. indicate an improvement in J_{sc} and comparable V_{oc} as CdS-only

devices by using a mixed CdS/CdSe layer. The work here shows that the use of just CdSe in place of CdS improves the J_{sc} but slightly negatively affects the Voc. So perhaps similar studies as performed here could be extended to devices with CdSe/CdS layers in the future.

References: appropriate credit to previous work?

As listed above there are a couple of very recent references that touch on similar observations for CdSe in CdTe devices. Perhaps the authors are not aware of these due to the recentness of the publications.

Clarity and context: lucidity of abstract/summary, appropriateness of abstract, introduction and conclusions

The data are generally clear in presentation, however there are a handful of minor points to address:

The scale bar in Figure 2 should indicate 5 μm , not 5 nm.

Page 13 - "TEM-SAD and SEM-SE imaging was used to verify that the large columnar grains adjacent to the FTO layer are zincblende for the 100 nm CdSe layer sample." Without further explanation, I'm unsure of how SEM-SE imaging could be used to verify the phase of these grains. Perhaps what is meant is that SEM imaging showed continuous columnar grains and SAD verified the phase.

Supplementary Materials, page 2 - "...while APT encompasses a much smaller FOV (a 0.34 μm tall by 0.015 μm wide cylindrical volume for Figure 5 in the main text)." The FOV for the APT data looks larger than 15 nm.

There are typos in the paper title, the caption for table 1, and a few other places in the text.

Reviewer #3 (Remarks to the Author):

As far as the solar cell efficiency results described in the paper are concerned they are not new; the results they show are similar to what Paudel already published in 2014. The novelty is that they did APT, TEM, EBIC investigation to understand the underlying reasons which make this paper interesting.

Some detailed comments:

The observation of "photoactive" and "not photoactive" Cd(Se,Te) depending on Se concentration is an interesting and important observation. The combination of EBIC and SAD patterns is a convincing study. It would be interesting to learn more about this issue. Could you find a phase diagram showing this transition or other related information?

The authors mention that they assume that First Solar is also using Se alloying. In the conclusion they mention that optimizing Se alloying can lead to 25% and more efficiency. However, The authors cells in this publication and APL 105, 183510 (2014) have low FF (<70%) and low efficiency <15%. Can the authors comment on where they think the difference between 15 and potentially 25% comes from? Do the investigations made allow to speculate about the reason for low performance? Authors have investigated lower efficiency cells than what they have published earlier. It would be interesting if they could perform these investigation on their high efficiency (15%) devices.

"...The increased VOC of the cell with the 100 nm CdSe layer may suggest that the junction is formed within the CdTexSe_{1-x} layer and not at the FTO/CdTexSe_{1-x} interface, but further experimental evidence must be gathered to validate this conclusion...."

I do not understand this thought. Higher Voc might indicate better surface passivation. Grading might lead to carrier separation similar to CIGS surface grading. Do you have evidence, that there is a change

from p to n-type Cd(Se,Te) within the front part ? Why should this indicate a buried junction ? Please explain it clearly.

The paper could be suitable for publication with major revision.

Response to Reviewer Comments:

We are very grateful for the reviewer's time in preparing a detailed and accurate review, which has certainly strengthened our manuscript. We have addressed all of the reviewer comments and a detailed response is provided below. For clarity, the reviewer comments are shown in black, our comments are shown in red, and changes made to the manuscript are shown in blue. A Microsoft Word document is attached with markups showing the changes we made to the manuscript. Two new figures, a new table, and discussion text were added to the supplementary materials section to increase clarity regarding the measurement procedures.

We are confident that we have addressed all of the reviewer concerns and we are certain you will find the manuscript acceptable for publication.

Reviewer #1:

The key results from this paper are that Se diffuses from a CdSe layer and forms a CdTe_{1-x}Se_x layer, which is photoactive. I do not feel that CdSe increases in J_{sc} can be considered a result from this paper as it this has already been published by Paudel in previous papers, particularly reference 3.

This paper is focused on using EBIC, TEM, and APT to understand the formation, crystalline structure, and properties of the CdTe_{1-x}Se_x layer in a functional CdTe device using a CdTe_{1-x}Se_x alloy layer to explain the J_{sc} increase from a microscopic perspective. The microscopy results show important information that will help improve the solar cell fabrication quality such that other photovoltaic parameters, such as Voc and FF, of cells using CdTe_{1-x}Se_x alloy layers can be improved. The increase in J_{sc} due to a CdTe_{1-x}Se_x layer is not a new result and we did not intend to advertise the increase in J_{sc} as a new result, however, we believe it is important to include the basic photovoltaic properties, such as the EQE and JV-curves, of the **EXACT** devices used in this study. Therefore, the reader can easily reference the bulk photovoltaic properties to microstructural changes induced by the CdTe_{1-x}Se_x layer.

We added the phrase: "*used in the microscopy studies*" to the first sentence in the results and discussion section to let the reader know that the EQE and JV curves are for the devices used in the microscopy studies. We added: "*as shown previously by Paudel et al.*" to the third sentence of the results and discussion section to let the reader know that the property measurements included in the manuscript are not new results and are included as a guide for the reader to directly compare with the microscopy results. Also, at the end of the first paragraph of the results and discussions section we included the following sentence to guide the reader to a more detailed study of the bulk solar cell properties:

"A more detailed review of the solar cell properties, including a statistical analysis, with respect to the CdSe thickness and the comparison of an optimized CdSe window layer cell to a control CdS window layer cell, which yielded similar power conversion efficiencies of nearly 15 %, can be found in Ref. 3."

This is in essence though a good piece of work. The microscopy and APT data is solid and EBIC analysis is interesting and correlates with the EQE analysis to explain the junction shift caused by thick CdSe layers. However I do not feel this is sufficient new data to warrant publication in a non-specific PV journal such as this. Furthermore, I feel the inability of this work to demonstrate an improvement in performance will limit the impact and interest in this work. The authors are very careful to phrase their discussion in terms of improving Jsc but this really is of little benefit if the Voc and FF levels cannot be at least maintained. As a result the CdSe incorporation has simply served to reduce efficiency. If increasing the Jsc alone is the sole aim why not simply use a smaller bandgap absorber to begin with? I understand the bandgap grading aims but until this can be demonstrated to improve performance rather than simply long wavelength response it is of little consequence. At a time when the majority of the research field are focussing on attempts to improve Voc, a process which actively decreases this will receive little uptake. The claim that first solar use a CdTe_{1-x}Se_x layer is also unsubstantiated and their bandgap grading (which maintains or improves Voc) could be achieved via a number of other routes.

The intentions of this work are not to show an overall efficiency performance increase over the world record (WR) cell; however, the work is impactful because CdTe_{1-x}Se_x technology must be used as a component in the world record (WR) cell, which has not been revealed by First Solar, but strong evidence of this is shown at the end of this response in green text. Also, Yang *et al.* (references 5 and 6 in the manuscript) have shown that CdTe solar cells fabricated using CdS/CdSe stacked window layers can produce higher efficiency CdTe solar cells than CdTe solar cells using a traditional CdS window layer. This work is also impactful because it shows the formation, crystalline structure, and properties of this very important CdTe_{1-x}Se_x alloy layer in a functional CdTe device from a microscopic perspective, which is the first observation of its kind. This information can be used to improve the CdTe_{1-x}Se_x alloy so that other parameters, such as the Voc and FF, can be improved by employing a CdTe_{1-x}Se_x alloy.

We agree that improvements of Jsc at this point in time will not drastically improve the efficiency of the final cell because the Jsc is close to the theoretical maximum. We expect that this in-depth microscopy study can provide new information to the growth community to optimize the CdTe_{1-x}Se_x alloy, such that the Voc and Jsc of CdTe cells using this technology can be improved.

Overall, the importance of this work is to identify the following:

- (1) The Jsc increase is caused by the CdTeSe alloy
- (2) The CdTeSe alloy is graded causing a band gap gradient
- (3) The formation of a zincblende alloy at the interface is critical
- (4) The alloy is non-uniform, which can explain the decreased Voc and FF

These provide guidelines for the future improvement of device performance.

We have added the following sentence to the end of the first paragraph in the introduction clarify these points to the reader:

"In addition, Yang et al. have shown that using CdS/CdSe window layer stacks in CdTe solar cells increases the efficiency compared to traditional CdS window layers.^[A,B] A detailed microscopic understanding in the growth, formation, and photoactivity of the CdTe_xSe_{1-x} alloy layer formed when CdTe is deposited on CdSe is necessary to optimize the fabrication of Se induced band gap graded CdTe solar cells to further increase solar property parameters, such as the Voc, fill factor (FF), and the overall efficiency of CdTe solar cells using Se diffusion."

We have also modified the fourth to last sentence at the end of the first paragraph in the introduction

to provide compelling evidence that the WR cell uses Se diffusion:

“A comparison of the external quantum efficiency (EQE) measurements of Paudel et al. to the world-record (WR) CdTe solar cell, together with the existence of a patent filed by First Solar (fabricator of the WR cell) suggest that band gap engineering using Se diffusion is one factor that enabled the Jsc increases of the WR cell over the past 4 years. [C-E]”

We also added this paragraph to the end of the discussion section:

“The best synthesized CdSe/CdTe solar cells using the same growth methods for the cells studied in this manuscript have had matched efficiencies to control CdS/CdTe solar cells (14.7 vs. 14.8 % for the best cells).³ The efficiency of the CdSe/CdTe solar cells can be improved by improving the Voc and FF, which were both lower than the CdS/CdTe control cells. Time-resolved photoluminescence (TRPL) measurements have shown significantly lower carrier lifetime as compared to non-alloyed CdTe thin films, indicating that increased recombination reduces the Voc and FF of the studied CdTe_{1-x}Se_x alloy layers. This may be due to the Se non-uniformity as shown by the APT results in Figure 5a. This occurs as the CdSe and CdTe layers react and Se diffuses from the CdSe layer into the CdTe and Te diffuses from the CdTe into the CdSe layer. Future works should aim to increase the Voc and FF by creating a more uniform graded CdTe_{1-x}Se_x alloy. Using a Se alloy source should improve the uniformity of the graded CdTe_{1-x}Se_x alloy, and further improve the device efficiency beyond a typical CdS/CdTe device. Also, the shorter minority carrier lifetimes of the CdSe/CdTe devices compared to the CdS/CdTe can be due to interfacial recombination at that FTO/CdTe_{1-x}Se_x alloy interface. Optimization of a buffer layer between the FTO and CdTe_{1-x}Se_x alloy layers, such as CdS, can further increase the device efficiency beyond CdTe solar cells that are not band gap graded, which has been shown by Yang et al. [A]”

Source: Solar Cell Efficiency Tables, Green et al.					
date	eff	Voc	Jsc	FF	Source
12-Oct	18.3	0.857	26.95	77	Solar Cell Efficiency Tables (Version 41) ^[F]
13-Jun	19.6	0.857	28.59	80	Solar Cell Efficiency Tables (Version 42) ^[G]
13-Dec	20.4	0.872	29.47	79.5	Solar Cell Efficiency Tables (Version 44) ^[H]
14-Aug	21	0.876	30.25	79.4	Solar Cell Efficiency Tables (Version 45) ^[I]
14-Dec	21.5	0.877	30.44	79.2	Solar Cell Efficiency Tables (Version 46) ^[D]
16-Feb	22.1	Not Published in SCET			http://www.nrel.gov/ncpv/images/efficiency_chart.jpg

Evidence that First Solar uses a graded CdTe_{1-x}Se_x alloy layer in the WR cell: (1) First Solar disclosed that they use a band gap graded cell in a talk at the 2015 Spring MRS meeting in San Francisco. (2) Figure 1 shows the EQE of the 18.3% WR device measured in October 2012, the 21.5% WR device fabricated in December 2014, and devices fabricated by the Toledo authors. The EQE of the 2014 WR cell (Figure 1a) and the Toledo CdSe/CdTe cell (Figure 1c) show an almost identical long wavelength response, while they both show clear differences in the long wavelength response compared to the October 2012 WR cell. Clearly, the WR solar cell EQE changes (due to band gap changes) from October 2012 to December 2014 are identical to those seen from a CdS window layer and a CdSe window layer in the Toledo facilities. (3) A US patent filed by First Solar (US20140373908) was published in December 2014 describing in detail an invention using CdSe and, more specifically, a CdTe_{1-x}Se_x alloy within a solar cell to improve device efficiency. We have attached this patent to the response, and the patent is referenced in the manuscript and can be easily accessed online

(<http://www.google.com/patents/US20140373908>). Here is a small excerpt from the document (line 0006): *“One embodiment is a photovoltaic device. The photovoltaic device includes a layer stack and an absorber layer including $CdSe_zTe_{1-z}$ is disposed on the layer stack, wherein “z” is a number in a range from about 0 to about 1.”*

Overall, these facts strongly infer that First Solar uses a $CdTe_{1-x}Se_x$ alloy in the WR solar cell.

Figure 1: The EQE measurements of the (a) October 2012 WR cell,^[F] (b) December 2014 WR cell,^[D] and (c) a window free, CdS, CdSe, and CdS/CdSe hybrid window layer solar cells.^[C]

Whilst the microstructural analysis provided is of a good quality some demonstration of bandgap engineering occurring, as invoked in the abstract, would be hugely beneficial to the work. If the authors could show they had produced a significant amount of bandgap grading through their structure this would be a more key finding.

The APT results shown in Figure 5 are direct evidence of band gap grading due to a graded Se alloy. Ref. 10 in the manuscript shows a detailed review of the band gap energy with respect to the composition in the $\text{CdTe}_x\text{Se}_{1-x}$ alloy. We have added the following sentences to the manuscript to emphasize this point:

“The 1D profiles clearly show that the $\text{CdTe}_x\text{Se}_{1-x}$ alloy for both samples has a graded Se and Te concentration with the Se rich region at the front of the cell. This is direct evidence that these solar cells are band gap graded because the band gap of the $\text{CdTe}_x\text{Se}_{1-x}$ alloy is dependent on the value of x , in which the band gap of CdTe can decrease by as much as ~8% when $x = 0.6$. A detailed review of the band gap energy with respect to the $\text{CdTe}_x\text{Se}_{1-x}$ alloy composition can be found in Ref. 10.”

Overall I feel this paper should be published in a more PV specific journal.

The paper is extremely well written and I find few typographical errors. The conclusions are appropriate and supported by the data although I disagree with glossing over the efficiency decreases to focus on J_{sc} . I have no concerns with the accuracy or validity of the data and all data analysis has been done to a high standard. The methodologies and processes employed are excellent as one would expect from groups of such high quality.

Thank you.

Specific corrections:

Abstract: Surely the Voc rather than J_{sc} is the key to breaking 25% efficiency? First solar are putting in little effort to increase J_{sc} and their bandgap grading has been done to improve the Voc.

We agree that the Voc and FF are the main parameters to break the 25% efficiency barrier, and we did not plan for the reader to interpret our abstract in such a way that the efficiency barrier can be broken by increasing the J_{sc} . We want to convey that understanding the $\text{CdTe}_x\text{Se}_{1-x}$ alloy layer is key to improve the layer, such that the Voc and FF of devices using this layer can be improved, hence increasing the efficiency. The argument is that the $\text{CdTe}_x\text{Se}_{1-x}$ alloy layer is an important component in the top of the line CdTe solar cell even though there is not much published research on the $\text{CdTe}_x\text{Se}_{1-x}$ alloy layer (mostly because of the proprietary nature of the WR cell). We also do not want to convey the message that this alloy layer alone can enhance the solar cell beyond 25%, as it is one component of many components in a high efficiency CdTe solar cell. This comment made us realize that we did not convey our thoughts properly, and we modified the second sentence of the abstract to read:

“A link between the formation, growth, composition, structure, and photoactivity of the $\text{CdTe}_x\text{Se}_{1-x}$ alloy is important to improve this layer, and efficiencies of solar cells using a graded $\text{CdTe}_x\text{Se}_{1-x}$ alloy.”

Figure 2a - Scale bars are missing, what is the colour scale bar units? Is this output current in pA or is EBIC current/Probe current?

The white lines in Figure 2a correspond to 5 μm , which we included in the Figure caption. We also included that the EBIC contrast corresponds to EBIC/Probe current in the Figure caption.

Figure 3 - Presumably this should be μm rather than nm ?

Absolutely, thanks for noticing this. We have changed the scale to μm .

Reviewer #2:

Summary of the key results

Using CdSe instead of CdS at the front contact increases the J_{sc} and wavelength absorption both for short and long wavelengths. This is due to the diffusion of Se into the CdTe creating zincblende $\text{CdTe}_{1-x}\text{Se}_x$. If the wurtzite structure of CdSe is retained, then the J_{sc} is diminished.

Originality and interest: if not novel, please give references

This is certainly of interest to those in the thin film solar cell community. Some of these claims are novel, some have recently been reported by others. The following references explore the use of CdSe in the window layer and all report improved J_{sc} and improved short and long wavelength performance. All of the references also all attribute the improvement to interdiffusion of the CdSe and CdTe to create a graded $\text{CdTe}_{1-x}\text{Se}_x$ layer:

Paudel, Naba R., and Yanfa Yan. "Enhancing the photo-currents of CdTe thin-film solar cells in both short and long wavelength regions." *Applied Physics Letters* 105 (2014): 183510.

Paudel, Naba R., et al. "Current Enhancement of CdTe-Based Solar Cells." *IEEE Journal of Photovoltaics*, 5 (2015): 1492-1496.

Yang, Xiaoyan, et al. "Preparation and characterization of pulsed laser deposited a novel CdS/CdSe composite window layer for CdTe thin film solar cell." *Applied Surface Science* 367 (2016): 480-484.

Yang, Xiaoyan, et al. "Preparation and characterization of pulsed laser deposited CdS/CdSe bi-layer films for CdTe solar cell application." *Materials Science in Semiconductor Processing* 48 (2016): 27-32.

The latter two explore the phase somewhat using XRD, but not on the detailed local scale that is presented here. The present work is novel in that it quantitatively explores the amount of interdiffusion that can occur and examines the local crystal structure of the $\text{CdTe}_{1-x}\text{Se}_x$ phase.

The first two references were included in the submitted manuscript, but we were unaware of the second two references as the manuscript was submitted in early 2016. Thank you for pointing out these references. We have read the papers by Yang *et al.*, who show that CdTe solar cells with a CdSe/CdS stack window layer have higher efficiencies than traditional CdS window layer samples. This result is very significant and strongly corroborates the motivation for performing the detailed microscopic study on the formation, structure, composition, and photoactivity of the $\text{CdTe}_x\text{Se}_{1-x}$ alloy after CdTe deposition on CdSe. We do not believe that the XRD results by Yang *et al.* are important to discuss in our manuscript because they are used to identify the crystalline quality of the CdSe/CdS thin films before the deposition of the CdTe, and do not provide details regarding the phase transformation of the $\text{CdTe}_x\text{Se}_{1-x}$ alloy during CdTe growth as our manuscript does. We have added the following sentences to the introduction to acknowledge this work and point the reader to the works by Yang *et al.*:

"In addition, Yang et al. have shown that using CdS/CdSe window layer stacks in CdTe solar cells increases the efficiency compared to traditional CdS window layers. ^[A,B]"

We have also added the two Yang *et al.* references to the end of the following sentence in the introduction:

“The solubility of Se into CdTe is much higher than S, and therefore, can easily diffuse from the CdSe layer into the CdTe during the CdTe growth and post-processing treatments. [A,B,J-M]”

We have also added the following sentence to the end of the results and discussion:

*“Optimization of a buffer layer between the FTO and CdTe_{1-x}Se_x alloy layers, such as CdS, can further increase the device efficiency beyond CdTe solar cells that are not band gap graded, which has been shown by Yang *et al.* [A]”*

Data & methodology: validity of approach, quality of data, quality of presentation

The use of electron diffraction, APT, and EBIC to examine the structure, composition, and photoactivity relationships of CdTe_{1-x}Se_x alloys are quite appropriate and together present an overall picture of what is occurring.

Thank you.

Appropriate use of statistics and treatment of uncertainties

The only mention of uncertainties is in the supplementary discussion of the APT data. While a counting statistics approach is one possible way of assessing the uncertainty, for APT it really represents the lowest possible uncertainty. A mention of the observed background levels, method of peak ranging, and any peak overlaps which require deconvolution would provide a more complete assessment of the uncertainty and reliability of the data.

This is a valid point. It is difficult to accurately quantify errors associated with background levels and peak overlaps for CdTe based solar cells due to the complexity of the mass spectrum. The fact that the Cd concentration in the APT dataset is close to 50% and does **not** vary as the Te and Se compositions vary is evidence that the error induced by background and peak overlaps is small. Also, the compositions of the CdTe_{1-x}Se_x alloy match well with the phase diagram as far as the compositions that can be wurtzite or zincblende, which was verified using TEM-SAD. We have added a Figure showing the mass spectrum and ranges used in the supplementary materials section as well as a table that shows the composition of an APT dataset using the manual decomposition (used in the line profiles and proximity histograms) and using the IVAS peak decomposition routine. The supplementary materials discussion, figure, and table are shown below:

“The mass spectrum associated with the APT dataset shown in Figure 5b with labeled ranges is displayed in Supplementary Figure 4. The following ions and their charge states have been ranged: Cd⁺, Cd²⁺, Te⁺, Te²⁺, Se⁺, Se²⁺, Te₂⁺, Te₂²⁺, Cd₂²⁺, Se₂⁺, Se₂²⁺, CdTe⁺, CdTe²⁺, CdSe⁺, CdSe²⁺, SeTe⁺, SeTe²⁺, Cd₂Te²⁺, Cd₂Se⁺, Cd₂Se²⁺, CdSe₂²⁺, CdTeSe⁺, Cd₂TeSe²⁺, Ga⁺ (to assess FIB damage), Cl⁺, CdCl⁺, and CdSeCl⁺. There exist several minor peak overlaps, but the major peaks accounting for most of the counted atoms do not have peak overlaps, and therefore, the minor peak overlaps for the major elements (Cd, Se, and Te) were ignored in the 1D line profiles and proximity histograms. The overlaps that were manually decomposed for the 1D line profiles are the Cd₂Se²⁺/CdCl and the CdTe/CdSeCl due to the low abundance of Cl. Supplementary Table 2 lists the calculated bulk compositions of an extracted region of the dataset displayed in Figure 5b with Cl enrichment (~2 M ions) using the peak decomposition routine embedded in IVAS 3.6.12 (CAMECA Instruments) and the manual decomposition used in the 1D line profiles and proximity histograms. The peak decomposition routine embedded in IVAS 3.6.12 includes a background subtraction routine, while background subtraction was not accounted for in the manual decomposition

for the 1D line profiles. Overall, the compositions using both methods are very similar, and the Cl concentration for the 1D line profile is increased by <0.1 at. % due to background. This is minor as the Cl segregation only occurs at the zincblend/wurtzite interface, in which the Cl content increases from 0 at.% to 2 at.% (figure 5 of the manuscript).

Supplementary Figure 4: The mass spectrum and associated ranges for the CdSe/CdTe solar cell APT dataset displayed in Figure 5b.

IVAS decomposition analysis			Manual Decomposition used in 1D line profile		
Ion	Decomposed Count	Atomic %	Ion	Decomposed Count	Atomic %
Cd	1171206.121	51.652%	Cd	1555603	50.774%
Se	651009.3252	28.711%	Se	907525	29.621%
Te	438935.9508	19.358%	Te	594617	19.408%
Cl	5982.566692	0.264%	Cl	9808.9166	0.320%
Ga	344.4053059	0.015%	Ga	461	0.015%

Supplementary Table 2. The obtained bulk compositions of the mass spectrum shown in Supplementary Figure 4 using the IVAS decomposition analysis and manual decomposition used in the 1D line profiles and proximity histograms.”

We also added the following sentences to the Figure 5 caption to guide the reader to the Supplementary Materials section for details regarding the mass spectrum:

“The error bars embedded in the 1D line profiles and proximity histograms are derived from counting statistics, while other minor uncertainties can occur from background levels and mass spectral peak

overlaps. A detailed summary of the mass spectrum, mass spectral ranges, and resulting compositions can be found in the supplementary materials section (Supplementary Figure 4 and Supplementary Table 2)."

Conclusions: robustness, validity, reliability

This presents a consistent microscopic picture of the observed changes in device performance and puts forth reasonable discussion regarding the results.

Suggested improvements: experiments, data for possible revision

The two references by Yang et al. indicate an improvement in J_{sc} and comparable V_{oc} as CdS-only devices by using a mixed CdS/CdSe layer. The work here shows that the use of just CdSe in place of CdS improves the J_{sc} but slightly negatively affects the V_{oc} . So perhaps similar studies as performed here could be extended to devices with CdSe/CdS layers in the future.

We are interested in performing similar experiments on CdSe/CdS layers in the future to correlate the composition and structure to the solar cell properties. The Toledo group has fabricated such cells and we plan to examine them in depth using the same microscopy techniques, which will be the focus of a future manuscript. Thank you for the suggestion.

References: appropriate credit to previous work?

As listed above there are a couple of very recent references that touch on similar observations for CdSe in CdTe devices. Perhaps the authors are not aware of these due to the recentness of the publications.

We added these references to the manuscript and appreciate the suggestion.

Clarity and context: lucidity of abstract/summary, appropriateness of abstract, introduction and conclusions

The data are generally clear in presentation, however there are a handful of minor points to address: The scale bar in Figure 2 should indicate 5 μm , not 5 nm.

This has been changed. Thank you.

Page 13 - "TEM-SAD and SEM-SE imaging was used to verify that the large columnar grains adjacent to the FTO layer are zincblende for the 100 nm CdSe layer sample." Without further explanation, I'm unsure of how SEM-SE imaging could be used to verify the phase of these grains. Perhaps what is meant is that SEM imaging showed continuous columnar grains and SAD verified the phase.

Thank you for pointing this out. You are correct. We have changed that sentence to:

"TEM-SAD was used to verify that the large columnar grains adjacent to the FTO layer shown in Supplementary Figure 1 are zincblende for the 100 nm CdSe layer sample."

Supplementary Materials, page 2 - "...while APT encompasses a much smaller FOV (a 0.34 μm tall by 0.015 μm wide cylindrical volume for Figure 5 in the main text)." The FOV for the APT data looks larger than 15 nm.

You are correct, we made an error. The field of view should be 50 nm in diameter. Thank you for noticing this typo. We have changed the text to 0.05 μm .

There are typos in the paper title, the caption for table 1, and a few other places in the text.

We identified the typo in the title. We changed Table 1 to read:

“Table 1 | Photovoltaic Performance Parameters. The Voc, Jsc, FF, and efficiency of the devices used in this study for the CdTe solar cells with different CdSe layer thicknesses.”

Reviewer #3:

As far as the solar cell efficiency results described in the paper are concerned they are not new; the results they show are similar to what Paudel already published in 2014. The novelty is that they did APT, TEM, EBIC investigation to understand the underlying reasons which make this paper interesting.

Correct. The aim of this paper is to correlate the photovoltaic properties of the structure, composition, and formation of the CdTe_xSe_{1-x} alloy for graded CdTe_xSe_{1-x} alloy CdTe solar cells.

Some detailed comments:

The observation of "photoactive" and "not photoactive" Cd(Se,Te) depending on Se concentration is an interesting and important observation. The combination of EBIC and SAD patterns is a convincing study. It would be interesting to learn more about this issue. Could you find a phase diagram showing this transition or other related information?

Yes, Reference 14 in the manuscript (Strauss *et al.*)^N includes the following phase diagram:

Fig. 1. Phase diagram of the CdTe-CdSe pseudobinary system

This phase diagram is only complete to 800C, while the growth of the CdTe layer on top of the CdSe layer was at a substrate temperature of 610C. We updated the phase diagram with the compositions and structures recorded from our measurements and added it to the Supplementary Materials section for reference. We removed the data from the two-phase region from Strauss *et al.* because the samples for Strauss *et al.* were annealed to an equilibrium state (~1 -30 days depending on the temperature), while our samples were grown on a 610C substrate for several minutes. We added the phrase in the parenthesis at the end of the second to last paragraph in the discussion section to direct the reader to the Supplementary Materials section for a phase diagram including the APT compositions and TEM-SAD structural data:

“Although both wurtzite and zinblende $CdTe_xSe_{1-x}$ structures can exist with 30% Se,^[0] it has been shown that the zinblende $CdTe_xSe_{1-x}$ alloy with ≤ 30 at.% Se results from using CSS growth techniques; this layer

is photoactive (a phase diagram with the APT compositions and TEM-SAD structures is shown in Supplementary Figure 5).^[N]

Also, the following has been added to the Supplementary Materials section:

“The structural and compositional data obtained from the TEM-SAD and APT measurements shown in Figure 3 and Figure 5 in the manuscript have been added to the phase diagram by Strauss et al. using the substrate temperature during the CdTe growth (610 °C). The updated phase diagram is shown in Supplementary Figure 5 with black and white dots representing the zincblende and wurtzite structures by Strauss et al. and the green and white dots representing the zincblende and wurtzite structures from the current work, respectively. The phase diagram by Strauss et al. was limited to a minimum temperature of 800 °C, and therefore, the phase diagram has been extended to the CdTe substrate growth temperature. The two phase data points that encompass a very small fraction of the phase diagram from Strauss et al. have been left out of the phase diagram to maintain simplicity. Strauss et al. annealed the specimens from 16 hours to 30 days, depending on the anneal temperature, to reach the equilibrium state, and then quenched the samples in water. The phases were identified using x-ray diffractometer patterns. On the other hand, the CdTe_xSe_{1-x} alloy used for the data points added to the phase diagram from this work was formed while CdTe was deposited on top of a thin CdSe layer at 610 °C for several minutes, followed by a 30 minute 390 °C CdCl₂ anneal, and a 30 minute 150 °C Cu diffusion. The specimen was not quenched after the anneal treatments.

Supplementary Figure 5. Phase diagram of the CdTe-CdSe pseudobinary system. The data have been obtained from previous work and current APT work.^[N] The inset depicts the unit cells of the photoactive zincblende structure and photo-inactive wurtzite structure, indicating a composition and structural dependence on photoactivity. The Cd sites are shown as gray, and the Te/Se sites are shown as yellow.”

The authors mention that they assume that Frist Solar is also using Se alloying. In the conclusion they mention that optimizing Se alloying can lead to 25% and more efficiency. However, The authors cells in this publication and APL 105, 183510 (2014) have low FF (<70%) and low efficiency <15%. Can the authors comment on where they think the difference between 15 and potentially 25% comes from? Do the investigations made allow to speculate about the reason for low performance?

We added the evidence supporting our conclusion that First Solar uses this technology in their solar cell, which is in green text below. The way we wrote “optimizing the CdTe_{1-x}Se_x alloy will lead to a 25% efficient solar cell” does not convey our thoughts properly. We meant to state that the CdTe_{1-x}Se_x alloy is one of many integral parts of the WR cell, and that the microscopic structure, formation, and properties of the CdTe_{1-x}Se_x alloy learned during the course of this study can help improve solar cells employing this technology. Optimizing this layer alone may not bring the efficiency to 25%, but will be one component in increasing the efficiency to an end goal of 25%. We have changed the last sentences of the conclusions to reflect this:

“A band gap graded absorber layer is one critical component for high efficiency CdTe-based solar cells. Therefore, the optimization of Se diffusion into the CdTe absorber layer is crucial for improving the state-of-the-art efficiency of CdTe based solar cells.”

We believe there is a large efficiency gap between our cell and 25% efficiency because there are other technologies besides the CdTe_{1-x}Se_x alloy included in the WR cell that are not part of the device stack used in the present study. Some components in the WR cell that also increase the efficiency that are not included in our cell include a Cu:ZnTe layer (revealed by First Solar at the 2015 Spring MRS Meeting), an anti-reflective coating, and most likely other technologies that have not been disclosed, which take major resources to optimize. The overarching goal of our paper is to reveal the link between nanoscale structure, composition, formation, and properties to bulk CdTe device properties using a CdTe_{1-x}Se_x layer.

More specifically for our devices, the low efficiency is mainly due to the low Voc and low FF. We speculate that the low Voc and low FF are due to two main reasons: (1) the Se alloy is not uniform as shown by the APT results in the manuscript. This is due to the fabrication process - Se diffusion from the CdSe window layer into the CdTe and Te diffusion from the CdTe into the CdSe at the reaction front. TRPL measurements showed much lower carrier lifetime as compared to the regular CdTe thin films. We expect from this study that a higher Voc and FF can be achieved using a Se alloy source to improve uniform Se alloy thin films instead of growing CdTe on top of CdSe. (2) The current buffer layer is not optimal for the Se alloy absorber. We expect high nonradiative recombination at the FTO/Se-alloy interface. We anticipate that a much higher efficiency can be achieved if these issues are solved. Unfortunately, current resources do not allow us to work and solve these issues. We hope that these results will stimulate other groups in PV community to work together to solve these issues.

We added the following paragraph to the end of the discussion to clarify issues to the reader and hopefully stimulate other CdTe growth groups to optimize the CdTe_{1-x}Se_x alloy layer:

“The best synthesized CdSe/CdTe solar cells using the same growth methods for the cells studied in this manuscript have had matched efficiencies to control CdS/CdTe solar cells (14.7 vs. 14.8 % for the best cells).^c The efficiency of the CdSe/CdTe solar cells can be improved by improving the Voc and FF, which were both lower than the CdS/CdTe control cells. Time-resolved photoluminescence (TRPL) measurements have shown significantly lower carrier lifetime as compared to non-alloyed CdTe thin films, indicating that increased recombination reduces the Voc and FF of the studied CdTe_{1-x}Se_x alloy layers. This may be due to the Se non-uniformity as shown by the APT results in Figure 5a. This occurs as the CdSe and CdTe layers react and Se diffuses from the CdSe layer into the CdTe and Te diffuses from the CdTe into the CdSe layer. Future works should aim to increase the Voc and FF by creating a more uniform graded CdTe_{1-x}Se_x alloy. Using a Se alloy source should improve the uniformity of the graded CdTe_{1-x}Se_x

alloy, and further improve the device efficiency beyond a typical CdS/CdTe device. Also, the shorter minority carrier lifetimes of the CdSe/CdTe devices compared to the CdS/CdTe can also be due to interfacial recombination at that FTO/CdTe_{1-x}Se_x alloy interface. Optimization of a buffer layer between the FTO and CdTe_{1-x}Se_x alloy layers, such as CdS, can further increase the device efficiency beyond CdTe solar cells that are not band gap graded, which has been shown by Yang et al.^[A]”

Authors have investigated lower efficiency cells than what they have published earlier. It would be interesting if they could perform these investigation on their high efficiency (15%) devices.

We agree that it would be very interesting to perform these measurements on the higher efficiency devices, and we plan on doing so in the future with more optimized CdTe_{1-x}Se_x alloy layers. However, the growth of the CdSe/CdTe solar cells used the same growth process as the 15% devices, and therefore, the formation, structure, and EBIC properties (photoactive or non-photoactive) of the CdTe_{1-x}Se_x alloy would not be different than the devices studied in this manuscript. We plan on performing similar measurements with optimized Se layers using a CdTe_{1-x}Se_x source as well as devices with an optimized buffer layer between the FTO and CdTe_{1-x}Se_x alloy when programmatic funding is available to produce such cells.

"...The increased VOC of the cell with the 100 nm CdSe layer may suggest that the junction is formed within the CdTexSe1-x layer and not at the FTO/CdTexSe1-x interface, but further experimental evidence must be gathered to validate this conclusion...."

I do not understand this thought. Higher Voc might indicate better surface passivation. Grading might lead to carrier separation similar to CIGS surface grading. Do you have evidence, that there is a change from p to n-type Cd(Se,Te) within the front part ? Why should this indicate a buried junction ? Please explain it clearly.

Our thought is that the increased Voc indicates that there is less recombination occurring in the active region, such as the pn-junction. Also, the EBIC measurements from the 200 nm and 400 nm devices clearly show that the pn-junction occurs a distance away from the FTO/CdTe_{1-x}Se_x interface. Therefore, the increased Voc may be due to the pn-junction occurring at a distance farther away from the FTO/CdTe_{1-x}Se_x interface, where less recombination would be expected to occur. We do not have experimental evidence for this presently, it is merely a hypothesis. Therefore, we think it is best to modify this sentence to reflect only the facts and not our hypothesis. Also, the 200 nm and 400 nm EBIC average line profiles show that the maximum EBIC current occurs within the CdTe_{1-x}Se_x layer, which is indicative of the maximum built-in field and junction position. We have changed the last two sentences of that paragraph to:

“The increased V_{OC} of the cell with the 100 nm CdSe layer compared to the 50 nm CdSe layer suggests that less recombination is occurring at the FTO/CdTe_xSe_{1-x} interface, such that the additional CdSe better passivates the surface between the FTO and CdTe_xSe_{1-x} layers. The EBIC ALPs clearly show that the junction is beyond the FTO/CdTe_xSe_{1-x} interface, i.e. within the CdTe_xSe_{1-x} layer, for the 200 and 400 nm samples, and that the final junction position (distance from the original FTO/CdSe interface) is dependent on the original CdSe layer thickness.”

The paper could be suitable for publication with major revision.

Reviewers' comments:

Reviewer #1 (Remarks to the Author):

The authors had convinced me they were correct about the CdSe inclusion in the record cell by their response, but this was followed by First Solar's announcement at the IEEE PVSC that they do indeed utilize CdSe based bandgap grading. The authors have clearly done an excellent job of reverse engineering and their clarifications added strengthen the paper significantly.

As I stated previously my primary concern with this work was not its quality but rather the interest the work would generate in the field, given the cell results seemed to indicate a decline in performance. However given the additional work identified by reviewer #2 and the FS announcement, my opinion on this has changed. There will now likely be a huge shift towards bandgap grading in CdTe with CdSe and as a result this work should be of significant interest. I therefore believe it is suitable for publication with a few small alterations.

There is still no clarification of the y-axis scale for Figure 3. Is this in nA, pA or is it arbitrary units? Is there any data for a CdS window cell for comparison (this could be referenced if already published)?

Time resolved PL data is mentioned to show reduced carrier lifetimes for the CdSe cells (p17 L338). Could this data be included? One of the keys to improved performance with the graded structure would be increased carrier lifetime. Understanding why the devices reported here show a decrease is quite an important result and makes the data worthy of inclusion. It would also be good to see how the variation in CdSe thickness impacts on the lifetime.

One of the clarifications made by first solar was that as well as Jsc improvements, the bandgap grading had a large impact on their high carrier lifetimes. I believe the authors could strengthen the motivation for this study by including this fact i.e. it is not solely about Jsc improvement but also potentially carrier lifetime (although not observed here). This is an optional comment though and I would leave this to the authors discretion.

Reviewer #2 (Remarks to the Author):

This paper is focused on using EBIC, TEM, and APT to understand the formation, crystalline structure, and properties of the CdTe_{1-x}Se_x layer in a functional CdTe device using a CdTe_{1-x}Se_x alloy layer to explain the Jsc increase from a microscopic perspective.

The suggested additional references, clarifications, and corrections have been made.

There is at least one fairly major peak overlap - Te₂₊₊ with Te⁺ - that is not mentioned in the new supplemental information. But the additional text, supplementary figure 4, and supplementary table 2 provide useful information regarding the data uncertainty.

Reviewer #3 (Remarks to the Author):

Authors have provided detailed explanations and appropriately revised the manuscript. The paper is suitable for publication.

Response to Reviewer Comments:

Reviewer #1:

The authors had convinced me they were correct about the CdSe inclusion in the record cell by their response, but this was followed by First Solar's announcement at the IEEE PVSC that they do indeed utilize CdSe based bandgap grading. The authors have clearly done an excellent job of reverse engineering and their clarifications added strengthen the paper significantly.

As I stated previously my primary concern with this work was not its quality but rather the interest the work would generate in the field, given the cell results seemed to indicate a decline in performance. However given the additional work identified by reviewer #2 and the FS announcement, my opinion on this has changed. There will now likely be a huge shift towards bandgap grading in CdTe with CdSe and as a result this work should be of significant interest. I therefore believe it is suitable for publication with a few small alterations.

There is still no clarification of the y-axis scale for Figure 3. Is this in nA, pA or is it arbitrary units? Is there any data for a CdS window cell for comparison (this could be referenced if already published)?

Author Response: The y-axis in the scale of Figure 3 is the measured EBIC current (the current measured from the cell) divided by the electron beam current. The division negates the Amps unit, and the value is therefore unitless. A value of 1 would indicate that there is no EBIC because the measured current at the cell would be equal to the current absorbed by the cell. In this case, the measured current would be the electron beam current. The electron beam is composed of high-energy electrons. For this experiment, the electrons had 3 kV each. For a band gap of ~ 1.4 eV (CdTe band gap), each 3 kV electron will produce approximately 600 carriers (the rule of thumb is that a carrier is generated when the electron loses 3 times the band gap energy). Therefore, EBIC/beam current cannot exceed 600 for a 3 kV accelerating voltage. I have referenced EBIC papers that clarify this point in the manuscript. I have also referenced a paper that shows the EBIC/probe current average line profile for a CdS/CdTe solar cell (Ref. 20).

Time resolved PL data is mentioned to show reduced carrier lifetimes for the CdSe cells (p17 L338). Could this data be included? One of the keys to improved performance with the graded structure would be increased carrier lifetime. Understanding why the devices reported here show a decrease is quite an important result and makes the data worthy of inclusion. It would also be good to see how the variation in CdSe thickness impacts on the lifetime.

Author Response: Yes, we have measured a shorter carrier lifetime for one CdSe/CdTe solar cell compared to a CdS/CdTe solar cell, but not for all of the CdSe thicknesses. We prefer not to publish this result because we would like to save it for a future publication. We do mention that our CdSe/CdTe solar cells produce shorter carrier lifetimes, and we also explain why this may be in the discussion. We hope to improve our devices in the future, and hopefully other groups can report devices with higher carrier lifetimes in the future. In the discussion we have written the following in reference to the low Voc and carrier lifetime of the CdSe/CdTe device:

"This may be due to the Se non-uniformity as shown by the APT results in Figure 5a. This occurs as the CdSe and CdTe layers react and Se diffuses from the CdSe layer into the CdTe and Te diffuses from the

CdTe into the CdSe layer. Future work should aim to increase the V_{oc} and FF by creating a more uniform graded $CdTe_{1-x}Se_x$ alloy."

One of the clarifications made by first solar was that as well as Jsc improvements, the bandgap grading had a large impact on their high carrier lifetimes. I believe the authors could strengthen the motivation for this study by including this fact i.e. it is not solely about Jsc improvement but also potentially carrier lifetime (although not observed here). This is an optional comment though and I would leave this to the authors discretion.

Author Response: This is a good point, and unfortunately we have not seen the same result for our cells at this point. Perhaps with some further optimization, we or other open science groups will be able to achieve better carrier lifetimes. I'm sure that First Solar has been optimizing the $CdTe_{1-x}Se_x$ alloy for a very long time, especially since their patent is from 2012, so there is a lot of work to do with the open science community. We would prefer to keep our argument for this particular article because we do not have any results that corroborate that Se alloying can improve the carrier lifetime. Hopefully, we can make this happen in the future and write about it.

Reviewer #2:

This paper is focused on using EBIC, TEM, and APT to understand the formation, crystalline structure, and properties of the $CdTe_{1-x}Se_x$ layer in a functional CdTe device using a $CdTe_{1-x}Se_x$ alloy layer to explain the Jsc increase from a microscopic perspective.

The suggested additional references, clarifications, and corrections have been made.

There is at least one fairly major peak overlap - Te_{2++} with $Te+$ - that is not mentioned in the new supplemental information. But the additional text, supplementary figure 4, and supplementary table 2 provide useful information regarding the data uncertainty.

Author Response: We have discussed the peak overlap with Te_{2++} and $Te+$ in the Supplementary materials. Although the peaks overlap quite a bit, the $Te+$ peaks are much larger than the Te_{2+} peaks, and do not contribute significantly to the counts.

Reviewer #3:

Authors have provided detailed explanations and appropriately revised the manuscript. The paper is suitable for publication.

Author Response: Thank you.

[REFERENCES FOR THE TRANSPARENT PEER REVIEW FILE:

- A. Yang, X. *et al.* Preparation and characterization of pulsed laser deposited CdS/CdSe bi-layer films for CdTe solar cell application. *Materials Science in Semiconductor Processing* **48**, 27–32 (2016).
- B. Yang, X. *et al.* Preparation and characterization of pulsed laser deposited a novel CdS/CdSe composite window layer for CdTe thin film solar cell. *Applied Surface Science* **367**, 480–484 (2016).

- C. Paudel, N. R. & Yan, Y. Enhancing the photo-currents of CdTe thin-film solar cells in both short and long wavelength regions. *Appl. Phys. Lett.* **105**, 183510 (2014).
- D. Green, M. A., Emery, K., Hishikawa, Y., Warta, W. & Dunlop, E. D. Solar cell efficiency tables (version 46). *Prog. Photovolt: Res. Appl.* **23**, 805–812 (2015).
- E. Duggal, A. R., Shiang, J. J., Huber, W. H. & Halverson, A. F. Photovoltaic Devices. (2013).
- F. Green, M. A., Emery, K., Hishikawa, Y., Warta, W. & Dunlop, E. D. Solar cell efficiency tables (version 41). *Prog. Photovolt: Res. Appl.* **21**, 1–11 (2012).
- G. Green, M. A., Emery, K., Hishikawa, Y., Warta, W. & Dunlop, E. D. Solar cell efficiency tables (version 42). *Prog. Photovolt: Res. Appl.* **21**, 827–837 (2013).
- H. Green, M. A., Emery, K., Hishikawa, Y., Warta, W. & Dunlop, E. D. Solar cell efficiency tables (version 44). *Prog. Photovolt: Res. Appl.* **22**, 701–710 (2014).
- I. Green, M. A., Emery, K., Hishikawa, Y., Warta, W. & Dunlop, E. D. Solar cell efficiency tables (Version 45). *Prog. Photovolt: Res. Appl.* **23**, 1–9 (2014).
- J. Wei, S.-H., Zhang, S. B. & Zunger, A. First-principles calculation of band offsets, optical bowings, and defects in CdS, CdSe, CdTe, and their alloys. *J. Appl. Phys.* **87**, 1304 (2000).
- K. McCandless, B. E., Hanket, G. M., Jensen, D. G. & Birkmire, R. W. Phase behavior in the CdTe–CdS pseudobinary system. *J. Vac. Sci. Technol. A* **20**, 1462–7 (2002).
- L. Ohata, K., Saraie, J. & Tanaka, T. Phase Diagram of the CdS–CdTe Pseudobinary System. *Jpn. J. Appl. Phys.* **12**, 1198–1204 (1973).
- M. Nunoue, S. Y., Hemmi, T. & Kato, E. Mass Spectrometric Study of the Phase Boundaries of the CdS - CdTe System. *J. Electrochem. Soc.* **137**, 1248–1251 (1990).
- N. Strauss, A. J. & Steininger, J. Phase Diagram of the CdTe–CdSe Pseudobinary System. *J. Electrochem. Soc.* **117**, 1420–1426 (1970).
- O. Sebastian, P. J. & Sivaramakrishnan, V. The growth and characterization of CdSe_xTe_{1-x} thin films. *J. Cryst. Growth* **112**, 421–426 (1991).]